# The Association of Variants within Types V and XI Collagen Genes with Knee Joint Laxity Measurements

**DOI:** 10.3390/genes13122359

**Published:** 2022-12-14

**Authors:** Samantha Beckley, Roopam Dey, Shaun Stinton, Willem van der Merwe, Thomas Branch, Alison V. September, Mike Posthumus, Malcolm Collins

**Affiliations:** 1Health through Physical Activity, Lifestyle and Sport Research Centre (HPALS) and the International Federation of Sports Medicine (FIMS) International Collaborating Centre of Sports Medicine, Division of Physiological Sciences, Department of Human Biology, University of Cape Town, Cape Town 7700, South Africa; 2Division of Biomedical Engineering and Division of Orthopaedic Surgery, Faculty of Health Sciences, University of Cape Town, Cape Town 7700, South Africa; 3End Range of Motion Improvement, Atlanta, GA 30324, USA; 4Sports Science Orthopaedic Clinic, Sports Science Institute of South Africa, Cape Town 7700, South Africa; 5Sports Science Institute of South Africa, Cape Town 7700, South Africa

**Keywords:** genu recurvatum, anterior-posterior tibial translation, external-internal tibial rotation, ligament, tendon, knee laxity

## Abstract

Joint laxity is a multifactorial phenotype with a heritable component. Mutations or common polymorphisms within the α1(V) (*COL5A1*)*,* α1(XI) (*COL11A1*) and α2(XI) (*COL11A2*) collagen genes have been reported or proposed to associate with joint hypermobility, range of motion and/or genu recurvatum. The aim of this study was to investigate whether polymorphisms within these collagen-encoding genes are associated with measurements of knee joint laxity and computed ligament length changes within the non-dominant leg. One hundred and six healthy participants were assessed for genu recurvatum (knee hyperextension), anterior-posterior tibial translation, external-internal tibial rotation and ligament length changes during knee rotation of their non-dominant leg. Participants were genotyped for *COL5A1* rs12722 (T/C), *COL11A1* rs3753841 (C/T), *COL11A1* rs1676486 (T/C) and *COL11A2* rs1799907 (A/T). The genotype-genotype combination of any two or more of the four *COL5A1* rs12722 CC, *COL11A1* rs3753841 CC, *COL11A1* rs1676486 TT and *COL11A2* rs1799907 AA genotypes was associated with decreased active and passive knee hyperextension. These genotype-genotype combinations, including sex (male), increased age and decreased body mass collectively, also contributed to decreased passive knee hyperextension. These findings suggest that *COL5A1*, *COL11A1* and *COL11A2* gene-gene interactions are associated with knee hyperextension measurements of the non-dominant leg of healthy individuals.

## 1. Introduction

Joint laxity, which is a multifactorial trait with a heritable component, is defined as the measured movement constrained by the structure of its ligaments, shape of bones and joint congruency [1,2]. Symptomatic joint hypermobility is a common clinical feature of classic Ehlers-Danlos syndrome (EDS), which is often caused by mutations within the α1(V) collagen gene (*COL5A1*) [3]. Furthermore, it was previously proposed that *COL5A1*, which encodes the α(1) chain of type V collagen, is associated with hypermobility spectrum disorders [4] and, although joint hypermobility is not a clinical feature, multifocal fibromuscular dysplasia (FMDMF) [5]. Common genetic variants within the *COL5A1* gene are also associated with several musculoskeletal soft tissue injuries [6], including anterior cruciate ligament (ACL) ruptures and other ligament injuries [7]. Interestingly, increased generalised joint hypermobility (GJH) and knee joint laxity are also intrinsic risk factors for ACL rupture [8].

*COL5A1* variants have been associated, albeit not extensively, with nonpathological joint laxity [9]. Specifically, the CC genotype of the *COL5A1* rs12722 T/C variant was associated with increased sit and reach range of motion, a measurement partially determined by joint laxity, in older healthy participants [10]. In addition, the rs12722 T allele was found to associate with increased genu recurvatum (knee hyperextension) and GJH in females [11].

Type XI collagen is a structural and functional homologue of type V collagen and its α(1)XI and α(2)XI chains are encoded by the α1(XI) (*COL11A1*) and α2(XI) (*COL11A2*) collagen genes, respectively [12]. Mutations within these genes cause Stickler Syndrome, which although not present in all cases, is also characterised by joint hypermobility, amongst other clinical features [13]. Pathogenic variants within these genes can also cause fibrochondrogenesis, nonsyndromic deafness, Marshall syndrome and/or otospondylomegaepiphyseal dysplasia (OSMED) [14]. Although the association with ACL rupture has not been investigated, variants within the domestic dog *COL11A1* genes have been reported to be associated with cranial cruciate ligament ruptures (analogous to the ACL in humans) [15]. The *COL11A1* rs3753841 T, *COL11A1* rs1676486 C and/or *COL11A2* rs1799907 T alleles are however associated with increased risk of other musculoskeletal soft tissue injuries [16,17]. Interestingly, these studies also reported that the *COL11A1* and/or *COL11A2* variants interacted with *COL5A1* variants to associate with risk of injury [16,17].

Although to our knowledge no studies have specifically investigated the association of the type XI collagen gene variants with any joint laxity measurements, we propose that genetic variants within *COL5A1, COL11A1* and/or *COL11A2* associate with the spectrum of measurements ranging from symptomatic joint hypermobility to the normal inter-individual variations in joint laxity [11,18]. An increase in joint laxity is also believed to be associated with an increased length in joint restraints, including the ligaments [19]. Therefore, larger changes in the length of the knee stabilising ligaments, including the ACL, posterior cruciate ligament (PCL), medial collateral ligament (MCL) and lateral collateral ligament (LCL) [20], may result in increased knee joint laxity.

This study aimed to investigate whether the *COL5A1* rs12722 C/T, *COL11A1* rs3753841 T/C, *COL11A1* rs1676486 C/T and/or *COL11A2* rs1799907 A/T variants are associated independently or via gene-gene interactions with genu recurvatum, anterior-posterior tibial translation (displacement) or external-internal tibial rotation of the non-dominant leg. An additional aim of the study was to determine whether these collagen gene variants were significant contributors in multiple linear regression models of the knee joint laxity measurements, as well as any computed knee ligament length changes within the non-dominant leg.

## 2. Materials and Methods

### 2.1. Participants

One-hundred and six moderately active (at least 150 min of moderate intensity exercise per week), apparently healthy participants were recruited from nearby fitness centres within the greater Cape Town area, as well as through social media and word of mouth, between February 2016 and October 2018. Participants were required to have no history of spinal cord injury or any history of injury to the non-dominant lower limb. An additional 28 participants with a history of injury (14 knee and 4 ankle) or treatment (*n* = 9) to the non-dominant lower limb were also recruited and included with the 106 participants in a sub-analysis (*n* = 134). Prior to testing, all participants were required to sign an informed consent form. Questionnaires detailing personal particulars, sporting activities, as well as previous medical and injury history, were completed by all participants. Since this was a genetic study, all participants included in the study were of self-reported European ancestry to minimise the possible effect of population stratification. Leg dominance was also self-reported and for the purpose of this study, data from the non-dominant leg was reported. This study was approved by the Human Research Ethics Committee of the Faculty of Health Sciences within the University of Cape Town (HREC ref: 859/2015).

Sit and reach assessments were measured as previously described [10]. Since this study was investigating inherent genotype effects, participants were instructed to refrain from performing any warm-up exercises prior to testing. This modification was adopted due to unpublished observations that range of motion (ROM) responds to external stimuli such as a warm-up in an individualised manner (Miller C-J et al., personal communication). The test was repeated and the larger of the two measurements was recorded. The Beighton Score was used to assess GJH as previously described [2].

### 2.2. Knee Joint Laxity Assessments

Passive and active genu recurvatum were measured in degrees as previously described using a goniometer while the participants were in the supine position on a plinth with a bolster under both feet. Lower degree values indicated increased knee hyperextension. For the active measurement, participants were instructed to hyperextend their leg [21].

The KT-1000 (MEDmetric Corporation, San Diego, CA, USA) arthrometer was used to assess anterior and posterior tibial translation, relative to the femur, at 133 N while the knee was positioned at 30° flexion. Active displacement was measured while the participant raised their foot in a controlled manner. Maximum displacement was recorded after the tibia was manually displaced by the tester until the endpoint.

Tibial external-internal rotation was measured using the Robotic Knee Testing (RKT) device as previously described [22]. The resulting data were used to produce load deformation curves, where the curve was divided into three sections: external rotation, area of play and internal rotation, which were defined by the turning points of the curve. External and internal rotation refers to the amount of rotation which occurred between the respective outermost point of the curve and turning points. Slack describes the amount of rotation which occurred between the two turning points of the curve (in the area of play) [22]. Reliability of the genu recurvatum and the anterior-posterior knee laxity measurements are summarised (Appendix A) and the reliability of the RKT measurements has previously been published [22].

### 2.3. Measuring Ligament Length Change

Absolute knee-stabilising ligament length changes were calculated from the RKT motion data. A previously validated musculoskeletal knee model with six degrees of freedom was used for this study [23]. The model was obtained from https://simtk.org/frs/?group_id=933 (accessed on 1 October 2019) and was visualised using OpenSim software (version 3.3) (Stanford University, Stanford, CA, USA). Flexion/extension, abduction/adduction, internal/external rotation and medial/lateral translation kinematic data obtained from the RKT testing were used in the model. Ligament length changes were determined from maximum external and internal rotation in four ligaments within the knee joint, separated into ten bundles. This included two anterior cruciate ligament bundles (aACL and pACL), two posterior cruciate ligament bundles (aPCL and pPCL), three medial collateral ligament bundles (aMCL, iMCL and pMCL), two deep layer bundles of MCL (aDMCL and pDMCL) and one lateral collateral ligament bundle (LCL). Absolute ligament length change was the difference between initial (at maximum external rotation) and final (maximum internal rotation) lengths of a ligament.

### 2.4. DNA Extraction and Genotyping

A 5 mL venous blood sample was collected from the forearm of participants and stored at −20 °C until DNA extraction. DNA was extracted as previously described and the DNA samples were stored at −20 °C [24]. DNA samples were genotyped for *COL5A1* rs12722 (C/T) using PCR and restriction fragment length polymorphism analysis as previously described [24]. The rs12722 T allele and TT genotype frequencies within the European population are 44.5 and 21.0%, respectively (www.ensemble.org; accessed on 4 November 2022).

TaqMan™ Genotyping Assays (Applied Biosystems, Waltham, MA, USA) were used for genotyping the DNA samples for *COL11A1* rs1676486 (T/C) and rs3753841 (C/T), as well as *COL11A2* rs1799907 (A/T), using the QuantStudio^®^ 3 Real-Time PCR System (Applied Biosystems) and Genotyping Application on the Thermo Fisher Cloud (https://apps.thermofisher.com/apps/spa/, (accessed on 4 November 2022)). All plates included negative controls containing no DNA as well as additional positive controls with known genotypes. All laboratory work was conducted at the Health through Physical Activity, Lifestyle and Sport Research Centre (HPALS) laboratories, Sports Science Institute of South Africa Building, Newlands, Cape Town. In the European population, the minor allele and genotype frequencies of rs3753841 is 38.8% C and 14.3% CC; rs1676486 is 17.0% T and 2.8% TT; and rs1799907 is 27.7% T and TT 8.9% (www.ensemble.org; accessed on 4 November 2022).

The TT and CT genotypes of *COL5A1* rs12722 were combined for all analyses because of the previous association of the injury protective CC genotype with ROM measurements [10,25]. The genotype profile combinations used in all analyses for rs3753841 (T/C), rs1676486 (C/T) and rs1799907 (T/A) were based on the previous association of the rs3753841 T, rs1676486 C and rs1799907 T allele-allele interactions with increased risk of Achilles tendinopathy [16] and the independent association of the rs3753841 TT genotype and the rs3753841 T and rs1676486 C allele-allele interactions with increased risk of carpal tunnel syndrome [17]. Similar to the *COL5A1* analysis, the *COL11A1* and *COL11A2* genotypes containing the allele associated with increased risk of musculoskeletal soft tissue injury were combined and compared to the protective CC, TT and AA genotypes for rs3753841, rs1676486 and rs1799907, respectively.

### 2.5. Statistical Analysis

The required sample size for this study was based on previously published research reporting the collagen genotype effects on knee joint laxity measurements [11]. The Shapiro-Wilk test was used to test for normality. Categorical data were compared using the Chi-squared or Fisher’s Exact tests. Unpaired *t*-test or one-way ANOVA were used for continuous normally distributed data, while Mann Whitney-U or Kruskal-Wallis tests were used for non-parametric data. Test for linear trend or Dunn’s multiple comparisons tests were used to compare groups.

Hardy-Weinberg Equilibrium was also calculated using R statistical software. The IBM SPSS Statistics software (version 25) or Prism (version 9.4.1) were used for all remaining statistical analyses. Significance was set at *p* < 0.05. Data were presented as mean and standard deviation if normally distributed, or as median and interquartile ranges (IQR) if not normally distributed.

Genotype scores were calculated as a proxy for genotype-genotype interactions and were based on the previous independent and/or collective association of *COL5A1* (rs12722), *COL11A1* (rs3753841 and rs1676486) and *COL11A2* (rs1799907) with musculoskeletal soft tissue injuries [16,17,26]. Each of the reduced injury risk *COL5A1* rs12722 CC, *COL11A1* rs3753841 CC, *COL11A1* rs1676486 TT and *COL11A2* rs1799907 AA genotypes were given a score of 2, such that participants with none, one, two, three or all four of the associated reduced injury risk genotypes had a total score of 0, 2, 4 or 8, respectively (none of the participants had a genotype score of 8).

To explore which physical characteristics or genotypes were significant predictors of knee joint laxity measurements, multiple linear regression analyses using backward elimination were performed. Assumptions of linear regression were tested.

## 3. Results

### 3.1. General Characteristics

The average age, height, body mass and BMI of the participants were 27.5 ± 5.4 years, 174.7 ± 9.9 cm, 72.3 ± 12.6 kg and 23.6 ± 3.0 kg/m^2^, respectively. Fifty-eight (54.7%) were male, while 91 (85.9%) reported being right leg dominant and 59 (55.7%) reported taking part in flexibility training. All genetic variants were in Hardy-Weinberg Equilibrium and allele as well as genotype frequencies were similar to those reported for European populations (*COL5A1* rs12722: *p* = 0.073, T allele = 56.2% and TT genotype = 26.9%; *COL11A1* rs3753841 *p* = 0.838, C allele = 39.2% and CC genotype = 16.0%; *COL11A1* rs1676486 *p* = 0.318, T allele = 17.9% and TT genotype = 4.7%; and *COL11A2* rs1799907 *p* = 0.653, T allele = 32.2% and TT genotype = 11.5%).

There were no genotype effects for *COL5A1* rs12722, *COL11A1* rs3753841, *COL11A1* rs1676486 or *COL11A2* rs1799907 on the general characteristics of the participants, their flexibility training, Beighton scores or sit and reach measurements (Appendix A). Similarly, there were no genotype effects for *COL5A1* rs12722, *COL11A1* rs3753841 or *COL11A2* rs1799907 on any of the general characteristics when the subset of participants tested using the RKT device were analysed. *COL11A1* rs1676486 could not be included in the sub-analysis because only three participants had the protective TT genotype (Appendix A).

### 3.2. Association of Collagen Genotypes with Knee Laxity Measurements

There was a significant (*p* = 0.017) decrease in the average external rotation measurement of the *COL5A1* rs12722 CC genotype group (4.6 ± 1.5°) compared to the combined TT and CT genotype group (5.6 ± 1.1°) (Appendix A). There were no other significant differences between the genu recurvatum, anterior-posterior tibial translation or external-internal tibial rotation measurements and the *COL5A1* (rs12722), *COL11A1* (rs3753841, rs1676486) or *COL11A2* (rs1799907) genotype groups in the non-dominant leg (Appendix A). Similar results were obtained when the 28 additional participants with a history of non-dominant lower limb injury and/or treatment were included in the analysis (Appendix A).

In order to investigate type V and XI collagen genotype interactions, genotype scores were calculated so that participants with none, any one, any two, any three or all four of the *COL5A1* rs12722 CC, *COL11A1* rs3753841 CC and/or rs1676486 TT and/or *COL11A2* rs1799907 AA genotypes had genotype scores of 0, 2, 4, 6 and 8, respectively. None of the participants had all four genotypes (genotype score of 8). Genotype scores of 4 (*n* = 13) and 6 (*n* = 4) were combined for the analysis. There were no genotype score effects on the general characteristics of the participants, their flexibility training, Beighton scores or sit and reach measurements when all the participants or only the subset of participants tested using the RKT device were analysed (Appendix A).

The average active genu recurvatum of the participants with a genotype score of 0 was lower (greater knee hyperextension) than that of participants with a genotype score of 4 or 6 (score 0: 173.4 ± 5.3°, *n* = 41; score 2 175.3 ± 6.1°, *n* = 44; score 4 or 6 176.7 ± 5.4°, *n* = 17; *p* for linear trend 0.029) (Figure 1A). Similarly, the median passive genu recurvatum of the 0 genotype score group was significantly lower (*p* = 0.048) that that of the 4 or 6 group [score 0: 176.0° (172.8°; 179.0°), *n* = 41; score 2: 177.0° (173.0°; 182.0°), *n* = 44; score 4 or 6: 179.0° (176.6°; 181.0°), *n* = 17] (Figure 1B). There were no genotype score effects on any of the anterior-posterior translation or external-internal rotation measurements of the non-dominant knee (Figure 2 and Appendix A). In support of this finding, there was a significant positive correlation between increase in genotype score and an increase in active (*r* = 0.245, *p* = 0.013) and passive (*r* = 0.217, *p* = 0.029) genu recurvatum of the non-dominant knee, but none of the other measurements. Similar results were obtained when the 28 additional participants with a history of non-dominant lower limb injury and/or treatment were included in the analysis (Appendix A). Interestingly, 17.3% (*n* = 9), 10.3% (*n* = 6) and 5.0% (*n* = 1) of the participants with genotype scores of 0, 2 and 4/6, respectively, were able to extend their non-dominant knee beyond 10° (hyperextension) during the Beighton test.

### 3.3. Contribution of Collagen Genotype Score to the Multiple Linear Regression Models

The *COL5A1*, *COL11A1* and *COL11A2* genotype score, as well as age, sex and body mass were significant predictors of measurements of passive genu recurvatum, and the model explained 18% of the variance compared to 15% when genotype was not included in the model (Table 1). Specifically, the analysis showed that being a male, having a one unit increase in age (year), a one kg decrease in body mass and a genotype score of 4 or 6 was predicted to decrease passive knee hyperextension by 5.00°, 0.23°, 0.13° and 3.55°, respectively, when all of the other variables were held constant. The genotype score was however not considered to be a significant predictor of active genu recurvatum (Appendix A). Similar results were obtained when the 28 additional participants with a history of non-dominant lower limb injury and/or treatment were included in the analysis (Appendix A).

### 3.4. Ligament Length Change

There were no significant differences in knee ligament length changes between the *COL5A1, COL11A1* and *COL11A2* genotype score groups (Appendix A).

## 4. Discussion

The main novel finding of this study was that participants with any two or more of the *COL5A1* rs12722 CC, *COL11A1* rs3753841 CC, *COL11A1* rs1676486 TT and *COL11A2* rs1799907 AA genotypes were significantly associated with reduced passive and active knee hyperextension of the non-dominant leg. Furthermore, these genotype combinations were also found to predict measurements of only passive knee hyperextension in a multiple linear regression model, where the effect size of the overall model was considered to be medium [27]. Additionally, the external tibial rotation was significantly lower in the *COL5A1* rs12722 CC genotype group. No other genetic associations were noted with any of the anterior-posterior tibial translation nor external-internal tibial rotation measurements.

The CC genotype of rs12722, which is located within the functional *COL5A1* 3′-UTR and is associated with increased mRNA degradation [28], was previously reported to be independently associated with increased sit and reach measurements in older, healthy participants of self-reported European ancestry [10], as well as with increased straight leg raise (SLR) measurements in young Asian college students [25]. This variant was however not associated with SLR measurements in Korean ballerinas [29] and Japanese athletes [30], as well as with whole body laxity [25]. Similar to previous studies, *COL5A1* rs12722 was not independently associated with sit and reach measurements in the younger participants included in this study [10,29,30]. The *COL5A1* rs12722 CC genotype was associated with increased extensible tendon structures for knee extensors, but not for plantar flexors [31], while the T allele was associated with greater measures of mechanical stiffness of the quadriceps muscle–tendon at moderate contraction levels [32]. It was however not associated with the volume, elastic modulus and *Z* of the patellar tendon [33]. Although not all studies have reported an association, several have reported an independent association of the *COL5A1* rs12722 CC genotype and/or C allele with a reduced risk (protective effect) of musculoskeletal soft tissue injuries, predominately tendinopathy (reviewed in [26]).

Types V and XI collagen are both fibril-forming collagens that are required to work in a coordinated manner to regulate nucleation and assembly of the collagen fibril within the developing tendon and other tissues [12]. Both *COL11A1* rs3753841 and rs1676486 are non-synonymous, resulting in substitutions of a leucine with a proline and a proline with a serine, respectively [34]. The rs1676486 T allele has also been reported to be associated with increased mRNA degradation [35]. The *COL11A2* rs1799907 variant produces different isoforms of the α2(XI) chain within the terminal acidic domain, an important binding site for a number of other molecules. It has been suggested that this alteration may affect the deposition of other collagen molecules on the fibril [36]. Since several variants have been shown to regulate mRNA degradation of type V and XI collagen mRNA, Hay et al. (2013) have proposed that these variants result in altered levels of types V and XI collagen production [16].

Compared to *COL5A1*, the potential biological effects of the *COL11A1* and *COL11A2* variants investigated in this study on musculoskeletal tissue function and properties have, to our knowledge, not been investigated. The *COL11A1* rs3753841 and/or *COL11A2* rs1799907 variants have however been associated with lumbar disc herniation [37], limbus vertebra in gymnasts [38], lumbar spine stenosis [39], ossification of the posterior longitudinal ligament of the spine [36] and rheumatoid arthritis [40]. In addition, interactions between the *COL11A1* rs3753841 T, *COL11A*1 rs1676486 C, *COL11A2* rs1799907 T and/or *COL5A1* rs7174644 AGGG alleles, the latter of which is also located within the functional *COL5A*1 3′-UTR, were previously reported to be associated with increased risk of Achilles tendinopathy in two populations [16] and carpal tunnel syndrome [17]. The reported combined association of the *COL5A1* rs12722 CC, *COL11A1* rs3753841 CC, *COL11A1* rs1676486 TT and *COL11A2* rs1799907 AA genotypes in this study aligns with the reports by several researchers that showed that reduced knee hyperextension is associated with a decreased risk of ACL rupture [41] and that the *COL5A1* rs12722 T allele was associated with increased knee hyperextension in females [11]. Future research should therefore evaluate these and other variants within the type XI collagen genes and the possible collective interactions with the *COL5A1* variants and their effect on musculoskeletal tissue function and properties.

While *COL5A1*, *COL11A1* and *COL11A2* genotype-genotype interactions were significantly associated with passive and active genu recurvatum, these genetic interactions only predicted passive genu recurvatum measurements. Passive range of motion measurements are more likely to represent underlying biological differences in musculoskeletal soft tissue mechanical properties when compared to active measurements, which may be additionally influenced by muscle size and strength. A limitation of the in-silico musculoskeletal knee model used to calculate ligament length changes (in OpenSim) was that certain subject-specific parameters could not be taken into account, such as bone geometry, joint space and joint capsule. The linear regression model presented in this study demonstrated the multifactorial nature of knee laxity as age, sex and body mass were all identified as predictors of genu recurvatum, but it also highlighted the contribution of specific genetic variants to knee laxity measurements. It is highly likely that the genetic contribution is polygenic. Since only four variants within three collagen genes were investigated in this study, future studies should investigate other genes involved in connective tissue biology. Moreover, the currently study had a relatively low sample size. Although associations were found, it is important that these associations are repeated in large cohorts. Since only a small number of participants included in this study tested positive for knee hyperextension, future research with a larger cohort may also investigate the association of these collagen genes within participants with symptomatic knee hyperextension and/or generalised joint hypermobility.

In conclusion, the findings of this study provide evidence supporting the role of variants within *COL5A1*, *COL11A1* and *COL11A2*, together with physiological characteristics age, sex and body mass in collectively associating with measurements of laxity, specifically passive genu recurvatum.

## Figures and Tables

**Figure 1 genes-13-02359-f001:**
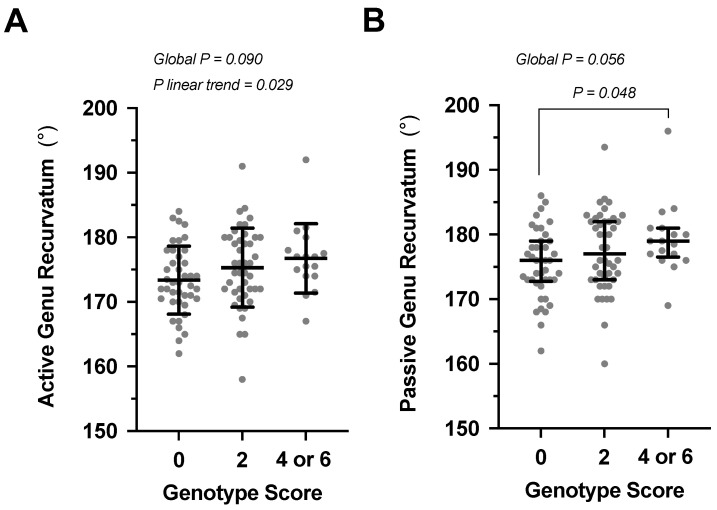
The (**A**) mean ± standard deviation of the active and (**B**) median (IQRs) of the passive genu recurvatum of the α1(V) (*COL5A1*), α1(XI) (*COL11A1*) and α2(XI) (*COLL12A1*) collagen genotype scores. Participants received a genotype score of 2 for each *COL5A1* rs12722 CC, *COL11A1* rs3753841 CC, *COL11A1* rs1676486 TT and *COL12A1* rs1799907 AA genotypes and 0 for all other genotypes, so that the genotype scores ranged from 0 to 8. None of the participants had a genotype score of 8 and the genotype scores of 4 and 6 were combined for the analysis. The individual values are shown as solid grey circles. The global test for linear trend and the Dunn’s multiple comparisons test *p* values are indicated. Lower degree values for genu recurvatum indicates a greater amount of knee hyperextension.

**Figure 2 genes-13-02359-f002:**
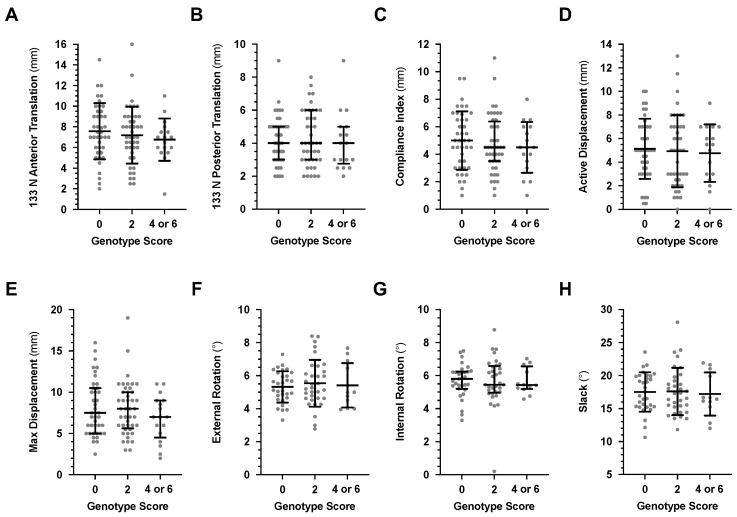
The mean ± standard deviation or median (IQRs) of the (**A**–**E**) five anterior-posterior laxity and (**F**–**H**) three external-internal rotation measurements of the protective α1(V) (*COL5A1*), α1(XI) (*COL11A1*) and α2(XI) (*COLL12A1*) collagen genotype scores. Participants received a genotype score of 2 for each *COL5A1* rs12722 CC, *COL11A1* rs3753841 CC, *COL11A1* rs1676486 TT and *COL12A1* rs1799907 AA genotypes and 0 for all other genotypes, so that the genotype scores ranged from 0 to 8. None of the participants had a genotype score of 8 and the genotype scores of 4 and 6 were combined for the analysis. The individual values are shown as solid grey circles.

**Table 1 genes-13-02359-t001:** Summary of multiple linear regression model for passive genu recurvatum in the non-dominant leg.

Passive Genu Recurvatum	Unstandardised Coefficient (β)	95% CI for β	SE β	*p*-Value
Constant	175.9	167.5 to 184.3	4.23	<0.001
Sex (male)	5.00	2.31 to 7.68	1.35	<0.001
Age (years)	0.23	0.03 to 0.44	0.10	0.025
Body Mass (kg)	−0.13	−0.24 to −0.02	0.05	0.018
Genotype Score (2)	1.56	−0.77 to 3.90	1.18	0.187
Genotype Score (4 or 6)	3.55	0.47 to 6.62	1.55	0.024
R^2^ = 0.22 and Adjusted R^2^ = 0.18				

CI = confidence interval; SE β = standard error of the coefficient; R^2^ = coefficient of determination. Units are shown in parentheses. The α1(V) (*COL5A1*), α1(XI) (*COL11A1*) and α2(XI) (*COL11A2*) collagen genotyping score coding: 0, 2 and 4 or 6. Participants received a genotype score of 2 for the *COL5A1* rs12722 CC, *COL11A1* rs3753841 CC, *COL11A1* rs1676486 TT and/or *COL11A1* rs1799907 AA genotypes and 0 for all other genotypes. Genotype scores, therefore, ranged from 0 (none of the injury protective genotypes) to 8 (all four injury protective genotypes). None of the participants had a genotype score of 8. An increase in the measurement of genu recurvatum in degrees indicates decrease in the amount of hyperextension.

## Data Availability

Not applicable.

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
