# Peer review of "The Association of Variants within Types V and XI Collagen Genes with Knee Joint Laxity Measurements"

_genes, 2022, doi:10.3390/genes13122359_

Round 1

Reviewer 1 Report

Reviews

1. Introduction: Authors mentioned that T-C-T haplotype was constructed from 2 SNPs of COL11A1 (Chr.1) and one from COL11A2 (Chr.6). I was not sure how a haplotype could be constructed from SNPs harboured in 2 distinct chromosomes. Either the reference (11) should be removed or the authors should come up with an explanation of the same.

2. All 106 participants were healthy individuals with no apparent history of joint mobility. It would be important to include at least a few patient samples. Further authors need to type the four SNPs COL11A1 rs3753841 T, COL11A1 rs1676486 C, COL11A2 340 rs1799907 T and/or COL5A1 rs7174644. Prevalence (or absence) of the genotypes in those patients will substantiate their findings.

3. For a population study where only 4 SNPs are typed, 106 control individuals are too low a number. This is further required as the statistical difference/contrast between the phenotypes correlated with the Genotype score is alarmingly low. It would be contentious to draw any clear inference from the study.

Author Response

We would like to thank the reviewer for reviewing the manuscript and their comments. We have revised the manuscript as follows:-

The following sections were requested to be improved:

  • Does the introduction provide sufficient background and include all relevant references?

Response

The introduction has been extensively rewritten.

  • Are all the cited references relevant to the research?

Response

We have check all references and revised where required.

  • Is the research design appropriate?

Response

The current study is designed to investigate phenotypic differences genotypes. This design is commonly used within genetic association studies.

  • Are the conclusions supported by the results?

Response

We have re-written the conclusions, specifically avoiding any over-interpretation with respect to injury mechanism

Comments and Suggestions for Authors

  1. Introduction: Authors mentioned that T-C-T haplotype was constructed from 2 SNPs of COL11A1(Chr.1) and one from COL11A2 (Chr.6)I was not sure how a haplotype could be constructed from SNPs harboured in 2 distinct chromosomes. Either the reference (11) should be removed or the authors should come up with an explanation of the same.

Response

This has been addressed in the revised introduction. The section now reads as follows:

“The COL11A1 rs3753841 T, COL11A1 rs1676486 C and/or COL11A2 rs1799907 T alleles are however associated with increased risk of other musculoskeletal soft tissue injuries [16, 17]. Interestingly, these studies also reported that the COL11A1 and/or COL11A2variants interacted with COL5A1 variants to associate with risk of injury [16, 17].”

  1. All 106 participants were healthy individuals with no apparent history of joint mobility. It would be important to include at least a few patient samples. Further authors need to type the four SNPs COL11A1 rs3753841 T, COL11A1 rs1676486 C, COL11A2 340 rs1799907 T and/or COL5A1 rs7174644. Prevalence (or absence) of the genotypes in those patients will substantiate their findings.

Response

The study design was to include only apparently healthy participants with no history of injury and/or treatment of the non-dominant lower limb (n=106). Identical results were obtained when 28 additional participants with a history of non-dominant lower limb injury and/or treatment were included in the analysis (Supplementary Tables S9 to S12 as well as S17 to S19). 

We agree with the reviewer that the association of these polymorphisms with symptomatic joint hypermobility should be investigated. However we believe that a meaningful analysis would require more than just a few patient samples and should therefore form part of future research. This has been included in the discussion as follows:

“Since only a small number of participants included in this study tested positive for knee hyperextension, future research with a larger cohort, may also investigate the association of these collagen genes within participants with symptomatic knee hyperextension and/or generalised joint hypermobility.” 

We did analyse the larger cohort included in this study and found that more participants with a genotype score of zero tested positive for knee hyperextension in the non-dominate leg. The following has been added to the results section:

“Interesting, 17.3% (n=9); 10.3% (n=6) and 5.0% (n=1) of the participants with genotype scores of 0, 2 and 4/6, respectively, were able to extend their non-dominant knee beyond 10° (hyperextension) during the Beighton test.” 

  1. For a population study where only 4 SNPs are typed, 106 control individuals are too low a number. This is further required as the statistical difference/contrast between the phenotypes correlated with the Genotype score is alarmingly low. It would be contentious to draw any clear inference from the study.

Response

The SNPs investigated in this study were all carefully selected, as discussed in the introduction, based off their previous associations with various musculoskeletal conditions and phenotypes. Previous studies have shown large effects of these variants in studies of similar sample size.

Reviewer 2 Report

It was a pleasure to review this very nicely written article. 

After carefully considering the data shown and the entire test, I thought several elements could contribute to increased relevance. 

Nomenclature issues: 

- row 43: 'benign joint hypermobility syndrome' is currently known as 'Hypermobility spectrum disorders' according to the 2017 classification of EDS; 

- row 45: the name of the syndrome is Stickler so maybe the text can be corrected;

- row 404: the correct term is 'polygenic' instead of 'polygenetic'

Pathogenic variants in these collagen genes are associated with a larger number of inherited disorder and I think this should be briefly mentioned, to give a better image of their role. 

It is not mentioned in this article how frequent these genotypes are in South-African population and how significant this small cohort is for this population. 

When providing dbSNP identifiers, it is also helpful to show the dbSNP release used to extract these. 

The measured effects are quite subtle and therefore, I think it would be very useful to have a larger cohort to support your observations and to increase significance. 

Given the complex nature of the musculoskeletal structure and function, the limited extent of the study does not seem to provide sufficient evidence to support relevant contribution to injury and I think this statement is not supported by the results. 

Author Response

We would like to thank the reviewer for their comments and suggestions which we used to revise and improve the manuscript.

The following sections could be improved:

  • Does the introduction provide sufficient background and include all relevant references?

Response

The introduction has been extensively rewritten.

  • Are all the cited references relevant to the research?

Response

We have check all references and revised where required.

  • Is the research design appropriate?

Response

The current study is designed to investigate phenotypic differences genotypes. This design is commonly used within genetic association studies.

  • Are the results clearly presented

Response

The results section has been revised

  • Are the conclusions supported by the results?

Response

We have re-written the conclusions, specifically avoiding any over-interpretation with respect to injury mechanism

Comments and Suggestions for Authors

It was a pleasure to review this very nicely written article. 

After carefully considering the data shown and the entire test, I thought several elements could contribute to increased relevance. 

Nomenclature issues: 

  • row 43: 'benign joint hypermobility syndrome' is currently known as 'Hypermobility spectrum disorders' according to the 2017 classification of EDS; 

Response

This has been corrected in the revised manuscript

  • row 45: the name of the syndrome is Stickler so maybe the text can be corrected;

Response

This has been corrected in the revised manuscript

  • row 404: the correct term is 'polygenic' instead of 'polygenetic'

Response

This has been corrected in the revised manuscript

  1. Pathogenic variants in these collagen genes are associated with a larger number of inherited disorder and I think this should be briefly mentioned, to give a better image of their role. 

Response

We have revised the introduction to include the following:

“Furthermore, it was previously proposed that COL5A1, which encodes the a(1) chain of type V collagen, is associated with hypermobility spectrum disorders [3] and, although joint hypermobility is not a clinical feature, multifocal fibromuscular dysplasia (FMDMF)”

“…….COL11A1 and COL11A2…… Pathogenic variants within these genes can also cause fibrochondrogenesis, nonsyndromic deafness, Marshall syndrome and/or otospondylomegaepiphyseal dysplasia (OSMED)”

  1. It is not mentioned in this article how frequent these genotypes are in South-African population and how significant this small cohort is for this population. 

Response

The following was in the Materials and Methods section (participants) “Since this was a genetic study, all participants included in the study were of self-reported European ancestry….”

The following were added to the Materials and Methods section (DNA extraction and Genotyping:

“The rs12722 T allele and TT genotype frequencies within the European population are 44.5 and 21.0% respectively (www.ensemble.org; accessed GRCh38.p13 on 4 November 2022).”

“The minor allele and genotype frequencies within the European population of rs3753841 is 38.8% C and 14.3% CC, rs1676486 is 17.0% T and 2.8% TT, and rs1799907 is 27.7% T and TT 8.9% (www.ensemble.org; accessed GRCh38.p13 on 4 November 2022).”

The following sentence in the Results section (General Characteristics) have also been modified as follows:

“All genetic variants were in Hardy-Weinberg Equilibrium and allele, as well as genotype,  frequencies were similar to those reported for European populations (COL5A1 rs12722: P = 0.073, T allele = 56.2% and TT genotype = 26.9 %;  COL11A1 rs3753841 P = 0.838, C allele = 39.2% and CC genotype = 16.0%; COL11A1 rs1676486 P = 0.318, T allele = 17.9% and TT genotype = 4.7% and COL11A2rs1799907 P = 0.653, T allele = 32.2% and TT genotype = 11.5%).”

Genotype counts were also added to Supplementary Tables S3 to S6.

  1. When providing dbSNP identifiers, it is also helpful to show the dbSNP release used to extract these.

Response

As indicated in the above response – these have been included.

  1. The measured effects are quite subtle and therefore, I think it would be very useful to have a larger cohort to support your observations and to increase significance. 

Response

The study design was to include only apparently healthy participants with no history of injury and/or treatment of the non-dominant lower limb (n=106). Identical results were obtained when 28 additional participants with a history of non-dominant lower limb injury and/or treatment were included in the analysis (Supplementary Tables S9 to S12 as well as S17 to S19). 

The following was added to the Materials and Methods:

“An additional 28 participants with a history of injury (14 knee and 4 ankle) or treatment (n = 9) to the non-dominant lower limb were also recruited and included with the 106 participants in a sub-analysis (n = 134).”

The following were added to the results:

“Similar results were obtained when the 28 additional participants with a history of non-dominant lower limb injury and/or treatment were included in the analysis (Supplementary Tables S9 to S12).”

Similar results were obtained when the 28 additional participants with a history of non-dominant lower limb injury and/or treatment were included in the analysis (Supplementary Tables S16 and S17).

Similar results were obtained when the 28 additional participants with a history of non-dominant lower limb injury and/or treatment were included in the analysis (Supplementary Table S18).

  1. Given the complex nature of the musculoskeletal structure and function, the limited extent of the study does not seem to provide sufficient evidence to support relevant contribution to injury and I think this statement is not supported by the results. 

Response

As indicated above, we have rewritten the manuscript to avoid any conclusions on mechanisms of injury, especial ACL rupture.

Round 2

Reviewer 1 Report

I find the revised version of the manuscript acceptable for publication in Genes. 

Reviewer 2 Report

Dear Authors,

It was a pleasure to review this version of your article "The association of variants within types V and XI collagen genes with knee joint laxity measurements". I thought this is an improved version of the article, with a detailed presentation of the genetic factors. I appreciated the extensive supplemental material.

However, I noticed an unusual term used on R52: 'Nonpathological joint laxity'. This is a term that, although used in some articles, is not generally accepted. Other terms like "isolated joint laxity" or "non-syndromic joint laxity" would be more appropriate.